# Repatriations of Ill and Injured Travelers and Emigrants to Switzerland: A Retrospective Analysis at a Tertiary Emergency Department from 2013–2018

**DOI:** 10.3390/ijerph18052777

**Published:** 2021-03-09

**Authors:** Lara Brockhus, Anne-Sophie Eich, Aristomenis Exadaktylos, Anne Jachmann, Jolanta Klukowska-Rötzler

**Affiliations:** Emergency Department, Bern University Hospital, 3010 Bern, Switzerland; lara.brockhus@outlook.com (L.B.); anne-sophie.eich@students.unibe.ch (A.-S.E.); aristomenis.exadaktylos@insel.ch (A.E.); Anne.Jachmann@insel.ch (A.J.)

**Keywords:** travel, repatriation, air transport, Rega, Switzerland

## Abstract

*Background*: As more and more people are travelling abroad, there are also increasing numbers who fall ill or have accidents in foreign countries. Some patients must be repatriated. While it has been reported that the number of repatriations is rising steadily, little is known about patients’ characteristics, calling for in depth investigations of this patient group. *Methods*: We have conducted a retrospective study including 447 patients repatriated to the Emergency Department at the University Hospital (Inselspital) in Bern, Switzerland from 2013–2018. *Results*: Between 2013 and 2018, the number of repatriated patients increased by 42.6%, from 54 to 77 cases. In total, 59% of these patients were male and the median age was 60 years. Overall, 79% of patients were repatriated from European countries, with the top five countries being Italy, France, Spain, Germany and Austria. About half the cases (51.9%) were caused by illness, the other half by accidents. In total, 127 patients had to undergo surgical intervention abroad; another 194 patients underwent surgery after repatriation. The hospitalization rate was 81.4%, with a median length of in-hospital stay of 9 days (IQR 5–14) at the Inselspital. The mortality rate of at the Inselspital hospitalized patients was 4.4%, with 16 patients dying within the first 30 days after repatriation. The median cost per case was 12,005.79 CHF (IQR 4717.66–24,462.79). A multiple regression analysis showed a significant association of total costs with hospitalization (*p* = 0.001), surgical intervention (*p* = 0.001), as well as treatment in the intensive care unit (*p* = 0.001). *Conclusions*: The number of repatriations has been continuously increasing in recent years and reached a mean value of more than one case per week at the Inselspital (77 cases per year in 2018). The 30 day-mortality rate of 4.4% and the median cost per case are relatively high, demonstrating a neglected Public Health concern. These findings may provide impetus—not only for further research into repatriations but also for Public Health Promotion strategies.

## 1. Introduction

World travel before the period of Coronavirus 2019 (COVID-19) was increasing, creating both new industries and new problems [1,2]. Travel is likely to continue post COVID-19 and according to UNWTO (World Tourism Organisation) forecasts international tourist arrivals will increase to 1.8 billion by 2030 [3].

Over the last decade, also Swiss citizens have tended to travel more. Domestic travel has remained stable, whereas the number of international journeys increased by 29.5% between 2012 and 2018, corresponding to 2.2 international journeys per person in 2018 [4]. Thereby, Tourism is increasing in all age groups, with increasing numbers of middle-aged and older people with preexisting medical conditions travelling abroad [5].

Most travelers return healthy and unharmed, but a few suffer from severe health problems abroad due to an injury or illness (partly from underlying medical conditions) [6,7]. These patients require medical attention and some are hospitalized abroad. On the basis of the patient’s state of health, the severity of the injury or illness as well as the quality of local health care facilities repatriation is planned [8,9].

If repatriation is necessary, and depending on the distance from Switzerland, patients are transported home either by a scheduled flight with or without medical assistance, by an ambulance jet or helicopter, by ground ambulance or privately. In Switzerland, most repatriations are provided by Swiss Air-Rescue—“Rega”—who conducted 1355 repatriations in 2018 to patients own or their relative’s place of residence [10].

Few studies have been published on the epidemiology of medical repatriation and patients’ characteristics and the majority of available studies is focusing on the description of cases, mortality and morbidity [11,12,13,14].

The study presented here draws on the particular conditions found in Switzerland and relies on a study conducted from 2000–2011 at the same major Swiss university hospital [15]. This study showed that the number of repatriations almost doubled in 12 years (21 patients in the 2000 to 41 in 2011) [15]. In the light of increasing numbers of travelers, it can be assumed that the number of repatriations will simultaneously continue to increase each year. The aim of this study was to analyze the collective/group of patients who were repatriated to the emergency department of a Swiss University hospital from January 2013 to December 2018. Thereby several aspects of these cases were analyzed, such as reason for repatriation, treatment and costs. Data were compared with pre-existing literature where available—focusing on the paper mentioned above.

## 2. Methods

### 2.1. Study Design

This single-center retrospective cross-sectional study was based on the demographic- and health-related data of 447 patients. We analyzed all patients over 16 years of age who were repatriated from foreign countries to the Inselspital emergency department in Bern from January 2013 to December 2018. Children under 16 years were not included, as they were admitted to a separate emergency department. No individual informed consent was obtained, yet patients who actively refused a general consent (GC = general consent) were excluded from the study (*n* = 31). Another 60 cases were excluded due to duplication of cases or incomplete data (cases with >50% missing data). Therefore, 447 cases were included in the study (Figure 1).

### 2.2. Data Collection and Extraction

All data were extracted and sorted anonymously from the routine records of the E.care digital data base system (E.care BVBA, ED 2.1.3.0, Turnhout, Belgium), the hospital information system (SAP) as well as the clinic information system (ipdos) from 1 January 2013 to 31 December 2018.

Relevant information included age, gender, profession, nature of the stay abroad, cause for repatriation, hospitalization abroad, surgical interventions abroad, means of transport to Switzerland, admission or transfer to other hospitals, duration of hospitalization at the Inselspital, operations performed, nature of discharge from the hospital, 30-day mortality and overall costs were also obtained.

For the patients’ professions, we used a classification system of a study on the same topic [15]: (a) retirement or disability pension, (b) social, health and education, (c) technical, (d) agriculture, building trade, and crafts, (e) state, justice and services, (f) economy, gastronomy and household, (g) language, journalism and art, (h) apprenticeship, unemployed and unknown.

The means of transportation were classified as followed (a) Rega (Swiss air rescue), (b) scheduled flight with/without accompanying medical care, (c) ground ambulance, (d) on their own, (e) unknown.

Patients in our emergency department are routinely triaged using the Swiss Emergency Triage Scale (STS) [16]. This triage system classifies the urgency of treatment for patients presenting to an emergency department in five levels: 1: acute life-threatening problem, 2: high urgency, 3: urgency, 4: less urgency, 5: non-urgent problem. This system is based on vital signs, which are collected during the triaging of patients (e.g., Glasgow Coma Scale, blood pressure, pulse), and determines the time period in which a patient should be treated [16]. Patients were triaged based on their clinical condition at the time of repatriation, not based on initial illness/injury. Data regarding diagnoses were collected from discharge reports for hospitalized patients or from emergency reports for outpatients and other transfers. They were then classified according to the previous study [15] for a better comparison.

### 2.3. Statistical Analysis

Categorical data were displayed in numbers and percentages. Continuous variables are displayed as means and standard deviations if normally distributed and as medians and interquartile ranges (IQR) if not normally distributed. The Mann–Whitney U test was used to compare groups. The total costs of hospitalization at the Inselspital were not normally distributed, so this variable was In-transformed before analysis. The effect sizes of the regression analyses are presented as geometrical mean ratios (GMR) and 95% confidence intervals. The GMR was obtained through exponentiation of the coefficients of a linear regression analysis, where the In-transformed outcome corresponds to the GMR of the outcome without In-transformation. A *p*-value < 0.05 was considered significant.

### 2.4. Ethical Considerations

The conduct of this descriptive study was approved by the cantonal ethics committee in Bern (No. 2019-02171). The study—including data extraction, anonymization, analysis, and storage—was performed in accordance with Swiss law, the standards of the local ethics committee and the Declaration of Helsinki [17].

## 3. Results

### 3.1. Study Population

Our study population included 447 patients who were repatriated to the Inselspital emergency department in Bern.

The overall number of repatriations increased over the period of our study, with most cases in 2017 (*n* = 94, 21%). With 54 cases in 2013 and 77 cases in 2018, there was an overall increase of 42.6%. There was a clear pattern in the distribution of the repatriations throughout the year; most cases occurred in late summer and early autumn (Figure 2).

About half of all patients were repatriated due to illness (*n* = 232, 51.9%) and the other half suffered an injury (*n* = 215, 48.1%).

In total, 58.8% (*n* = 263) were male. The median age was 60 years (IQR 44–71), and 37.1% of all patients were more than 65 years old (*n* = 166).

Eight patients were in a critical condition on arrival (urgent triage category according to our triaging system or high staffing levels expected) and therefore initially treated in our ER intensive care room. We identified five incidents that required repatriation of more than one person (2–3 people). All these incidents were due to traffic accidents abroad, involving multiple patients. There were no accumulations of repatriations due to natural disasters or accidents and no cases of primary evacuations (for example from accidents). Patients either presented themselves to an emergency department or other kind of doctor’s office (i.e., general physician, orthopedic surgeon) abroad.

91.5% (*n* = 409) of the patients were hospitalized abroad before repatriation, mostly for under a week (*n* = 266, 64.9%). Only 6.6% of patients were hospitalized abroad for more than 3 weeks (*n* = 27). A total of 127 patients (28.4%) underwent surgery abroad (Table 1).

Patients were repatriated from a total of 74 countries worldwide. Overall, 78.8% of patients were repatriated from European countries (*n* = 352) and 8.1% from Asian countries (*n* = 36). The top five countries patients were repatriated from were Italy, Spain, France, Germany and Austria (Table 2). Figure 3 gives an overview of the countries with the percentage of patients that were repatriated from each country.

### 3.2. Illnesses

Among the patients with illness, almost a fifth suffered from a cerebrovascular insult (*n* = 42, 18%). The second largest group of patients had diseases of the gastrointestinal tract (including the liver), followed by patients with primary infections (Table 3).

### 3.3. Injuries

The number of patients with monotrauma (*n* = 94, 43.7%) or multiple injuries (*n* = 99, 46.1%) were similar. Only 22 patients suffered from polytrauma (*n* = 22, 10.2%). In cases of monotrauma, the largest group of patients suffered an injury to the lower limbs (*n* = 41), followed by injuries of the back (*n* = 21) (Table 3).

Overall, 81.5% of the patients younger than 35 years suffered an injury. There is a significant relationship between age-group and type of injury (*X*^2^ = 31.42, *p* < 0.001), where the type of injury (traffic, sports, fall and others) depends on the age group (<50 years and ≥50 years of age). While younger people were more likely to be involved in traffic or sports accidents, injuries in older people were more often caused by falls.

### 3.4. Hospitalisation

Of all 447 patients, 365 (81.7%) were hospitalized at the Inselspital after repatriation (medically *n* = 167, surgically *n* = 198), 17 (3.8%) were directly transferred to another clinic and 65 (14.5%) were treated as outpatients. The median length of hospitalization at the Inselspital was 9 days (IQR 5–14). In total, 194 of the hospitalized patients underwent surgery; the median length of hospitalization for these patients was 11 days (IQR 7–17), whilst patients who did not undergo surgery stayed in the hospital for 7 days (IQR 4–10). For patients who were treated in the intensive care unit, the median length of stay was 10 days (IQR 5–19). In general, patients over 65 years stayed for a median of 9 days (IQR 6–14), while patients younger than 65 years stayed for 8 days (IQR 5–14).

### 3.5. Mortality

We analyzed all patients who were repatriated to the Inselspital and admitted to our hospital or transferred to other clinics. Sixteen of these patients (4.4%) died within the first 30 days. Most deaths were caused by deterioration of an existing pathology or their complications (e.g., gastrointestinal bleeding, metastasis, sepsis). Their median age was 65 years (IQR 57–75) (Table 3). Eleven of these patients were treated at the intensive care unit (68.8%) and four patients died within the first 24 h after repatriation.

### 3.6. Costs

The median cost per case was 12,005.79 Swiss Francs (CHF) (IQR 4717.66–24,462.79).

Table 4 shows the univariable associations of studied variables and the total costs. Hospitalization leads to a significant (*p* = 0.001) increase in the geometric mean of the total costs by 11.66 (95% CI 9.41–14.44) compared to outpatient treatment. A multiple regression (Table 5) including all variables, showed significant associations of total costs with (i) hospitalization (GMR 6.19, 95% CI 5.03–7.62, *p*: 0.001), (ii) having surgery (GMR 2.81, 95% CI 2.37–3.33, *p*: 0.001), (iii) being treated at an intensive care unit (GMR 1.79, 95% CI 1.40–2.30, *p*: 0.001) and (iv) being admitted as medical or surgical patient (GMR 0.79, 95% CI 0.68–0.93, *p*: 0.005).

## 4. Discussion

### 4.1. Trend

In the study period of 6 years, there was an increase in the number of repatriated patients from 54 in 2013 to 77 in 2018 (with a peak of 94 in 2017). This equals an increase of 42.6%. In the same period of time, statistics on travel behavior of Swiss citizens showed an increase of 29.5% in the number of itineraries abroad [4,18]. The number of repatriations from whole of Switzerland increased by 35.2% between 2013 and 2018 [18]. With numbers of travels abroad constantly rising, it is foreseeable that there will also be an increasing number of repatriations. Consequently, the increase in the amount of repatriations can only partly be explained by the increase of itineraries abroad by Swiss citizens. Since the Inselspital is one of the largest hospitals in Switzerland and a University hospital of maximum care, relevantly contributing to the medical care of repatriated patients (with 3 helipads for Rega helicopters to approach), more serious cases are assigned to it, which could explain the higher numbers in repatriation cases in comparison to whole Switzerland.

Most patients were repatriated from European countries, which reflects the travel patterns of Swiss people. In 2018, 89.8% of Swiss travel destinations abroad were European countries [4]. The differences in the healthcare systems and quality of care between the European countries combined with the unknown fact, if the repatriated patient came from a rural or an urban place, prohibits the drawing of conclusions regarding the underlying reasons for the repatriation. Additionally, people who travel to European and mostly neighboring countries probably wish for medical treatment back in Switzerland due to the proximity.

### 4.2. Illness and Injuries

About half (51.9%) of all patients had to be repatriated due to illness, the other half due to injuries (48.1%). This distribution appears to be stable in comparison to the study conducted from 2000 to 2011 [15].

Neurological diseases (included strokes, bleedings, tumors and other neurologically related illnesses) were one of the main medical reasons. In our study, cerebrovascular strokes were the most common disease with 18/16.8% and are of special importance, as early treatment is an essential factor for the outcome of the patient [19]. Age distribution and causes for injuries were similar to the previous study as well.

### 4.3. Hospitalisation

The rate of hospitalization (81.7%) in this population is fairly high compared to other cases at the Inselspital emergency department. The median rate of hospitalization between 2013–2018 was 32.75% (factor 2.47).

The fact that over 90% of all repatriated patients had already been hospitalized abroad can be a sign for a higher level of severity of the illness or injury. Moreover, repatriation becomes necessary in severe cases, which allegedly will need further in-hospital treatment. This is also evident in the median length of hospitalization of 9 days.

### 4.4. Mortality

The 30-day mortality rate of 4.4% in the hospitalized repatriated patients is very high. However, patients who need to be repatriated are more frequently in a critical condition. In comparison to the study from 2000 to 2011, the 30-day mortality rate remained stable [15]. As most patients were repatriated from European countries, where medical care can be provided, patients might only need repatriation when critical conditions occur. Additionally, when repatriating older patients, pre-existing comorbidities leading to more complex problems have to be taken into account. Most deaths were caused by deterioration of an existing pathology or resulting complications (e.g., gastrointestinal bleeding, metastasis, sepsis). As a 2009 study from the Netherlands showed, up to 82% of repatriations were necessary due to acute complications of a preexisting chronic disease [5].

### 4.5. Costs

Another aspect included in the study were the costs generated by the cases. The median cost per case was 12,005.79 CHF, with a median cost of 1374.89 CHF for outpatient treatment and 15,220.11 CHF for inpatient treatment.

We compared different groups of patients and identified parameters that had an impact on the case-costs: surgery (*p* = 0.001), hospitalization (*p* = 0.001), admission as medical/surgical patient (*p* = 0.001) and treatment in the intensive care unit (*p* = 0.001). The age or 30-day mortality of a patient did not significantly influence the case-costs after controlling for confounders in a multiple regression analysis. The effect of age on the total hospital costs, which appeared in a univariable linear regression analysis, disappears in a multiple regression analysis after controlling for confounders. This suggests that the parameter of age alone does not cause higher costs, but that older patients undergoing surgery or being treated in an intensive care unit generate greater costs.

We were not able to find any preexisting data on costs generated by repatriations in other hospitals.

#### 4.5.1. Strengths and Limitations

An important strength of this study was the analysis of a large number of repatriated patients, for whom we obtained detailed information. On the plus side, the retrospectively study approach at the course of hospitalization provided us with definite discharge diagnoses. Additionally, this is the first study to analyse the overall costs generated by the cases.

Our results could be compared to a previous study, conducted from 2000–2011 at the same hospital, which is one of the biggest university hospitals in Switzerland. However, from a national or global perspective this regionality of only one center might be a weakness.

#### 4.5.2. Outlook

Unfortunately, we could not collect complete data regarding comorbidities due to outpatient treatments and secondary transfers. The relevance of these comorbidities on patient outcome therefore remains unknown. Additionally, further follow-up data should be collected to obtain more comprehensive results. It would be interesting to know which reasons exactly lead to repatriation (better medical care, fear of insufficient medical care, basic aspiration of hospitalization in the home country). A study by Greuters et al. demonstrates that repatriations of ill or injured patients should always be sought when possible. However, the medical condition, the benefit for the patient and the materials required for transport must always be taken into account [5].

Regarding the sparse international literature, our study could inspire further epidemiological analyses. Additional follow-up of patients should be done to better asses the outcome of cases.

#### 4.5.3. Relevance of This Paper

We see different reasons why this study is relevant. Firstly, the rate of surgical interventions is very high in this patient population (43.4%). Many of the patients had already undergone surgery abroad. Surgery always involves a certain risk for the patients, which is why the indication for surgery should be examined particularly carefully in repatriated patients. Furthermore, the mortality rate in this population is very high, which is why they need special consideration. Lastly, but not least, the number of repatriations is continuously increasing. In 2018, with 77 cases per year, the average number of repatriations was 1.5 cases per week. Repatriations are becoming a regular occurrence, so larger and more comprehensive studies should be conducted.

## 5. Conclusions

The number of repatriations has been continuously increasing in recent years, with the largest group of patients being elderly Swiss nationals repatriated from neighboring countries. The causes were equally distributed between illness and injury with a median cost about 12,000 CHF per case. We found a high rate of surgical interventions, with almost half of all patients undergoing surgery after repatriation. Many of these had already undergone surgery abroad.

As numbers of repatriations are rising, with more than one case per week on average, the relevance of this topic is constantly increasing. All in all, the 30 day-mortality of 4% was relatively high, so that special consideration for repatriated patients is needed.

Compared to a study previously conducted at our emergency department [15], similar results were found. In particular, there was a further constant increase in the number of cases. Since the last study, still very little literature has been published regarding this topic.

These findings may provide impetus—not only for further research into repatriations but also for Public Health Promotion strategies, including advice for possible preventive measures at pre-travel-consultations.

## Figures and Tables

**Figure 1 ijerph-18-02777-f001:**
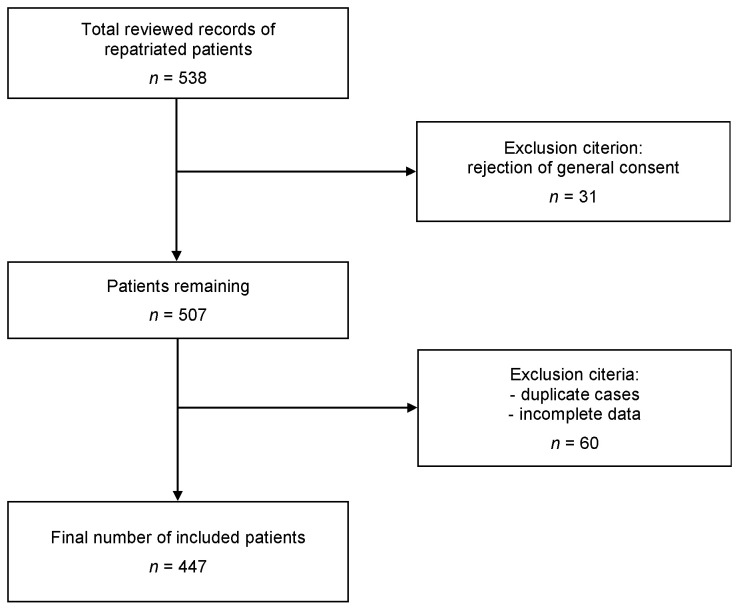
Flow chart of medical record selection.

**Figure 2 ijerph-18-02777-f002:**
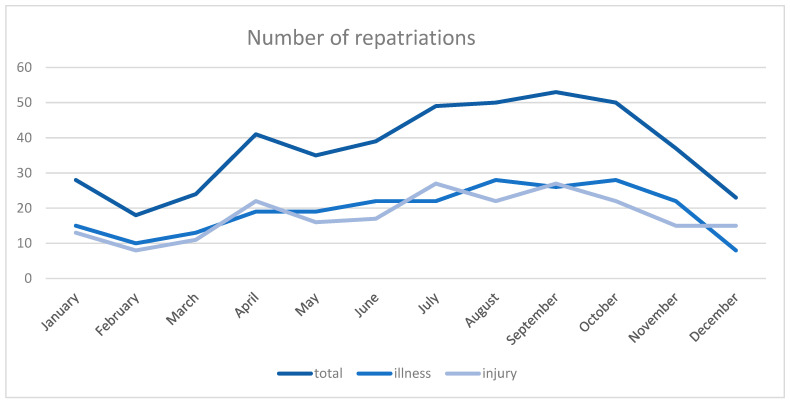
Cumulated number of repatriations by month for the years 2013–2018.

**Figure 3 ijerph-18-02777-f003:**
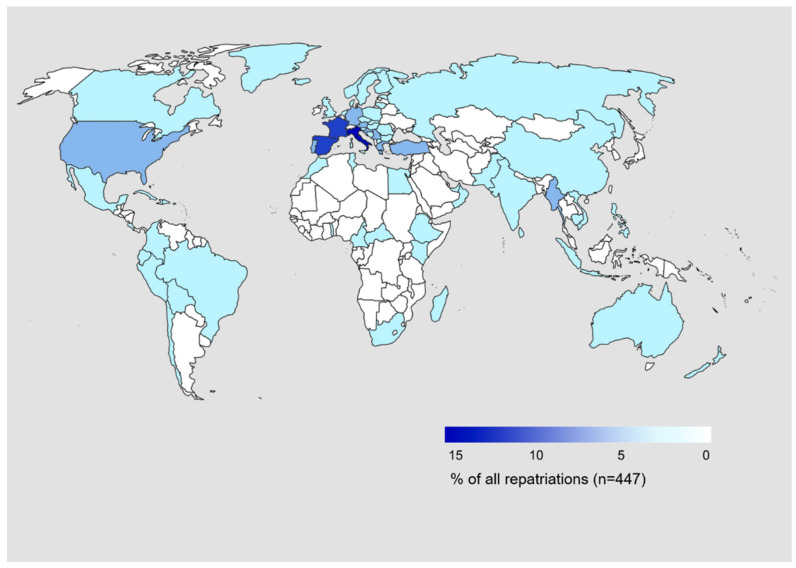
Illustration of repatriation countries by percentage.

**Table 1 ijerph-18-02777-t001:** Overview results.

Category	Cases	%
Years		
2013	54	12.1
2014	69	15.4
2015	78	17.4
2016	75	16.8
2017	94	21.0
2018	77	17.2
Gender		
Male	263	58.8
Female	184	41.2
Age		
>65 years	166	37.1
<65 years	281	62.9
Care abroad		
Hospitalisation	409	91.5
<7 days	266	
1–3 weeks	116	
>3 weeks	27	
Presentation GP/ED	27	6.0
Missing data	11	2.5
Mean of transport		
Rega (aeromedical transport)	289	64.7
Scheduled flight	59	13.2
Ground ambulance	73	16.3
Private	11	2.5
Unknown	15	3.4

**Table 2 ijerph-18-02777-t002:** Numbers and percentages of top 5 countries.

Countries	Cases	%
Italy	82	18.3
Spain	66	14.8
France	53	11.9
Germany	21	4.7
Austria	19	4.3

**Table 3 ijerph-18-02777-t003:** Overview diagnoses and 30-day mortality.

Group	Cases (*n*)	Percentage (%)	Male (%)	Median Age (Years) (IQR)
Illness	232	51.9	58.2	65 (55–73)
Nervous system	102	44	52.9	64 (54–71)
ischemic cerebrovascular insult	42	41.2	69	70 (62–74)
tumours	20	19.6	55	62 (59–69)
other	17	16.7	23.5	43 (40–62)
bleeding	12	11.8	50	60 (56–69)
epilepsy	10	9.8	30	63 (53–69)
infections	1	0.9	100	61
Cardiovascular system	29	12.5	58.6	66 (50–74)
myocardial infarction	10	34.5	70	71 (66–74)
other	17	58.6	47	57 (47–73)
angina pectoris	2	6.9	100	68 (64–71)
Infections	27	11.6	70.4	66 (56–71)
primary	23	85.2	69.6	67 (59–74)
secondary	4	14.8	75	57 (52–61)
Gastrointestinal system (incl. liver)	24	10.3	58.3	61 (57–68)
Respiratory system	22	9.4	68.2	72 (56–75)
Kidneys and urinary tract	9	3.9	88.9	69 (61–72)
Psychiatry	8	3.4	62.5	75 (60–78)
Musculoskeletal system	5	2.2	40	72 (70–76)
Haematological system	2	0.9	50	72 (71–74)
Rheumatology	2	0.9	100	52 (42–63)
Eyes, ears and nose	1	0.4	0	62
Endocrinology	1	0.4	0	84
Injury	215	48.1	59.5	52 (35–65)
Monotrauma	94	43.7	52.1	56 (33–70)
Head	14	14.9	71.4	47 (26–71)
Back	21	22.3	52.4	48 (29–59)
Thorax	2	2.1	0	49 (46–52)
Abdomen	0	-	-	-
Becken	2	2.1	50	44 (31–57)
Upper Extremities	14	14.9	42.9	64 (56–74)
Lower Extremities	41	43.6	51.2	57 (40–70)
Multiple injuries	99	46.1	61.6	51 (37–62)
Polytrauma	22	10.2	86.4	51 (37–62)
Cause				
Car	25	11.6	52	48 (31–63)
Bicycle	28	13.0	82.1	58 (44–62)
Motorbike	41	19.1	75.6	36 (26–56)
Sport	14	6.5	78.6	38 (24–51)
Water	10	4.7	50	37 (31–57)
Air Crash	12	5.6	75	35 (31–50)
Fall Bodyheight	54	25.1	31.5	70 (59–76)
Fall Higher	20	9.3	60	53 (28–67)
Violence	5	2.3	100	43 (31–45)
Other	6	2.8	33.3	45 (36–67)
30-day mortality	16	4.4	68.8	65 (57–75)
Illness	13	81.3	69.2	61 (57–75)
Neurological pathologies	6	46.2	83.3	67 (58–75)
Respiratory conditions	3	23.1	66.6	75 (72–76)
Gastrointestinal diseases	2	15.4	50	59 (58–60)
Secondary postoperative Infection	2	7.7	50	57 (56–58)
Injury	3	18.8	66.7	72 (62–78)
Motorbike accident	1	33.3	0	51
Fall from body height	1	33.3	100	83
Unidentified head injury	1	33.3	100	72

**Table 4 ijerph-18-02777-t004:** The association of outcome parameters with total case costs (univariate linear regression). Results of log transformed linear regression, expressed as geometric mean ratio.

Parameter	Median Costs (ICR)	Geometric Mean Ratio (95% CI)	*p*-Value
Age			
≥65	13,264.56 (6431–26,735)	1.33 (1.04–1.71)	**0.025 ***
<65	11,051.07 (3562–22,881)		
Hospitalisation			
yes	15,220.11 (8854–29,293)	11.66 (9.41–14.44)	**0.001 ***
no	1374.89 (986–1885)		
Intensive Care			
yes	24,770.28 (11,318–48,290)	3.05 (2.11–4.41)	**0.001 ***
no	11,030.75 (3383–21,844)		
Admission			
Injury	13,784.96 (3928–24,336)	0.93 (0.73–1.19)	0.556
Illness	10,995.34 (5506–24,463)		
Surgery			
yes	22,881.49 (15,091–39,610)	4.83 (3.96–5.88)	**0.001 ***
no	6596.11 (1791–11,127)		
Mortality 30-day			
yes	14,934.26 (7073–25,466)	1.37 (0.71–2.65)	0.341
no	11,499.65 (4532–24,197)		
Triage			
1–2 (urgent)	14,204.39 (7819–32,479)	1.40 (1.01–1.95)	**0.047 ***
3–5 (less urgent)	11,194.20 (4230–23,768)		

* Significant level ≤ 0.05.

**Table 5 ijerph-18-02777-t005:** Multivariable regression model of the association between total hospital costs with different parameters. Results of log transformed linear regression, expressed as geometric mean ratio.

Parameter	Geometric Mean Ratio (95% CI)	*p*-Value
Age	1.11 (0.95–1.30)	0.187
Hospitalisation	6.19 (5.03–7.62)	**0.001 ***
Intensive Care	1.79 (1.40–2.30)	**0.001 ***
Admission (injury vs. illness)	0.79 (0.68–0.93)	**0.005 ***
Surgery	2.81 (2.37–3.33)	**0.001 ***
30-day Mortality	0.71 (0.47–1.06)	0.095
Triage	1.00 (0.82–1.22)	0.965

* Significant level ≤ 0.05.

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
