# Peer review of "Repatriations of Ill and Injured Travelers and Emigrants to Switzerland: A Retrospective Analysis at a Tertiary Emergency Department from 2013–2018"

_ijerph, 2021, doi:10.3390/ijerph18052777_

Round 1

Reviewer 1 Report

Thank you for the opportunity to review this article. I have enjoyed reading this paper. Please see some suggestions, which are aimed at making the paper more translatable to the wider international air ambulance/ aeromedical community.

Well done on a very good paper, and I hope my comments are helpful.

Introduction

  • The opening sentence of “World travel is increasing…” reads oddly in the current COVID-19 environment, were travel has basically stopped. While I understand this is pre-COVID-19, I would recommend revising. The following paper discusses how COVID-19 has effected retrievals. You may wish to amend this sentence to reflect COVID-19 and the make the statement that travel is likely to continue post-vaccination.
    • Gardiner FW, Gillam M, Churilov L, Sharma P, Steere M, Hannan M, et al. Aeromedical retrieval diagnostic trends during a period of Coronavirus 2019 lockdown. Internal medicine journal. 2020.
  • This sentence is not accurate “Little has been published on the epidemiology of medical repatriation….” Internationally there has been much published, including recent papers from Norway and Australia. Search “Air Ambulance” and “Aeromedical”.

Methods

  • In the introduction, last paragraph, you write ED and then in methods and throughout you write emergency department.
  • I would like more information of on the service itself, including crew mix during aeromedical retrieval. Also I am not clear if your aeromedical retrievals are via helicopter or aircraft.
  • How were your in-flight diagnosis coded? Or did you use the final ED discharge diagnosis. These two items can be different, especially for neurology conditions such as ‘suspected stroke’. Or did you use the original referring hospital to code the diagnosis?
  • While you have the STS described, you do not appear to report severity by diagnosis or demographics. This would be interesting and may highlight interesting results. See recent research by the Royal Flying Doctor Service in Australia.
  • I understand the use of the term “repatriated” as you are returning people back to their home country, however did you do any primary evacuations (say during a motor vehicle accident) in addition to your inter-hospital transfers? I actually feel the terms primary evacuations and inter-hospital transfer (or secondary retrieval) reflect the true nature of the paper. Regionally in Australia and Brittan the term repatriation is use for low acuity patients being taken home after a primary evacuation and treatment.

Results

  • Much of your results could be discussed (in the discussion) in an international context reflective of the wider aeromedical literature. The mean ages and diagnostics trends appear to be similar with other published literature.
  • You abbreviate ER. In the first instance please define, i.e. emergency room.
  • As mentioned, would it be possible to provide more detailed diagnostic information in a table. You may consider using ICD-10 coding, see: https://icd.who.int/browse10/2016/en
  • This sentence “Figure 3 gives an overview of the countries with the percentage of patients that were repatriated from each country or island” needs two be reworded. For example Australia is a country, island, and continent. Also Tasmania is part of Australia, or did you only do retrievals from the Australian mainland?
  • Figure 3, could you add numerical values by country? Or region? Over each area within the figure, this would help make it more applicable by area.
  • Can we also have information on distance travelled, in addition to crew mix?
  • Section 3.2 “foreign countries which aren’t” can you amend to are not.
  • Table 3. I do not find this table very useful as it lacks detail. For example under the heading ‘Nervous system: cerebrovascular insult’, how many of these were hemorrhagic or ischemic strokes? Given the older cohort age, I would have assumed many of the 42 cases would be stroke. This would make it more relatable to other aeromedical services publications, such as:
    • Gardiner FW, Bishop L, Dos Santos A, Sharma P, Easton D, Quinlan F, et al. Aeromedical retrieval for stroke in Australia. Cerebrovascular Diseases. 2020;49(3):334-40.
  • Section 3.3. I would also be of interest to have the breakdown of injury type, such as head trauma. This would make it more relatable to other aeromedical services publications.
  • Other than mortality was any other follow-up conducted?
  • Cost section. There has been some aeromedical cost literature from Australia on mental health retrievals, however I agree the literature is lacking. This is a further strength of your paper.
  • Do you have severity codes? 1=high severity 2=moderate 3= low . Or does the triage categories cover clinical severity?
  • Would be interesting to have more information on diagnosis by severity, age, and gender, and mortality rates.

Discussion

  • What does UNZ mean?
  • I think much of this section requires more attention to the wider international literature. Specifically relating your findings to similar findings internationally.
  • Statements like “people who travel to European and mostly neighbouring countries probably wish for medical treatment back in Switzerland due to the proximity. As the Swiss population is used to a very high quality of medical care, the fear of low quality and/or wrong treatment might be higher”, needs to be referenced, or acknowledged as a topic for future research. Did you collect any patient views on this?
  • Mortality section. You mention pre-existing comorbidities. Did you collect this information? Would be a very useful inclusion in the results.
  • Some literature from Australia.

Author Response

Author's Reply to the Review Report (Reviewer 1)

Thank you for the opportunity to review this article. I have enjoyed reading this paper. Please see some suggestions, which are aimed at making the paper more translatable to the wider international air ambulance/ aeromedical community.

Well done on a very good paper, and I hope my comments are helpful.

Introduction

  • The opening sentence of “World travel is increasing…” reads oddly in the current COVID-19 environment, were travel has basically stopped. While I understand this is pre-COVID-19, I would recommend revising. The following paper discusses how COVID-19 has effected retrievals. You may wish to amend this sentence to reflect COVID-19 and the make the statement that travel is likely to continue post-vaccination.
  • Gardiner FW, Gillam M, Churilov L, Sharma P, Steere M, Hannan M, et al. Aeromedical retrieval diagnostic trends during a period of Coronavirus 2019 lockdown. Internal medicine journal. 2020.

The COVID-19 pandemic has been mentioned with a link to your suggested paper (85-87).

  • This sentence is not accurate “Little has been published on the epidemiology of medical repatriation….” Internationally there has been much published, including recent papers from Norway and Australia. Search “Air Ambulance” and “Aeromedical”.

Methods

We would like to refer to our statement that much has been published regarding aeromedical primary evacuations (nationally) – we do however not discuss this topic in our study.

  • In the introduction, last paragraph, you write ED and then in methods and throughout you write emergency department.

All abbrevations of ED/ER/UNZ have been changed to “emergency department” for clarity (121, 172, 174, 204, 420).

  • I would like more information of on the service itself, including crew mix during aeromedical retrieval. Also I am not clear if your aeromedical retrievals are via helicopter or aircraft.
  • How were your in-flight diagnosis coded? Or did you use the final ED discharge diagnosis. These two items can be different, especially for neurology conditions such as ‘suspected stroke’. Or did you use the original referring hospital to code the diagnosis?

We used the final emergency department (for outpatient or transferred treatment) or hospitalization diagnoses. Therefore, no initial diagnoses were analysed.

  • While you have the STS described, you do not appear to report severity by diagnosis or demographics. This would be interesting and may highlight interesting results. See recent research by the Royal Flying Doctor Service in Australia.

We have reviewed the Royal Flying Doctor Service literature, but here they analyse primary evacuations. Unfortunately, this evaluation could not be applied to our patient population. Since most patients were hospitalised abroad, we did not treat initial illnesses or injuries.

  • I understand the use of the term “repatriated” as you are returning people back to their home country, however did you do any primary evacuations (say during a motor vehicle accident) in addition to your inter-hospital transfers? I actually feel the terms primary evacuations and inter-hospital transfer (or secondary retrieval) reflect the true nature of the paper. Regionally in Australia and Brittan the term repatriation is use for low acuity patients being taken home after a primary evacuation and treatment.

Our definition of repatriation - in the sense of re (back) and patria (home) - was used in the same way in other papers, such as in

  • Sussman NM. Repatriation transitions: Psychological preparedness, cultural identity, and attributions among American managers. Int J Intercult Relations. 2001
  • Kraimer M, Bolino M, Mead B. Themes in Expatriate and Repatriate Research over Four Decades: What Do We Know and What Do We Still Need to Learn? Annu. Rev. Organ. Psychol. Organ. Behav. 2016

We also like to add that no primary rescue operations were carried out.

Results

  • Much of your results could be discussed (in the discussion) in an international context reflective of the wider aeromedical literature. The mean ages and diagnostics trends appear to be similar with other published literature.

As we elaborate in our statement, unfortunately very little international literature was found on our specific topic. Where possible, comparisons were always made with the existing literature.

  • You abbreviate ER. In the first instance please define, i.e. emergency room.

All abbrevations of ED/ER/UNZ have been changed to “emergency department” for clarity (121, 172, 174, 204, 420).

  • As mentioned, would it be possible to provide more detailed diagnostic information in a table. You may consider using ICD-10 coding, see: https://icd.who.int/browse10/2016/en

Our classification is based on the following study:

  • Hasler RM, Albrecht S, Exadaktylos AK, Albrecht R. Repatriations and 28-day mortality of ill and injured travellers: 12 Years of experience in a Swiss emergency department. Swiss Med Wkly 2015;145(November):1–9

which took place at our ED from 2000-2011 in Bern in order to make better comparisons over time.

  • This sentence “Figure 3 gives an overview of the countries with the percentage of patients that were repatriated from each country or island” needs two be reworded. For example Australia is a country, island, and continent. Also Tasmania is part of Australia, or did you only do retrievals from the Australian mainland?
  • Figure 3, could you add numerical values by country? Or region? Over each area within the figure, this would help make it more applicable by area.

This was a very helpful suggestion. To include the numbers within the figure made it very confusing. Instead, we have added a table with the Top 5 countries including the percentages for more clarity (255/256, 258-276).

  • Can we also have information on distance travelled, in addition to crew mix?

Unfortunately, we only had data at our disposal regarding the country from which the patients were repatriated rather than the exact city/hospital. Neither do we have information regarding the exact route, they took. Especially in large countries (e.g. Australia) it can become quite inaccurate to guess the exact route under these circumstances.

  • Section 3.2 “foreign countries which aren’t” can you amend to are not.

To shorten the results section, this sentence was removed completely. We checked the complete paper for other accidental abbreviations.

  • Table 3. I do not find this table very useful as it lacks detail. For example under the heading ‘Nervous system: cerebrovascular insult’, how many of these were hemorrhagic or ischemic strokes? Given the older cohort age, I would have assumed many of the 42 cases would be stroke. This would make it more relatable to other aeromedical services publications, such as:
  • Gardiner FW, Bishop L, Dos Santos A, Sharma P, Easton D, Quinlan F, et al. Aeromedical retrieval for stroke in Australia. Cerebrovascular Diseases. 2020;49(3):334-40.

We have revised the table, adding more details in order to describe the study population.

As you suspected “cerebrovascular insults” describe only ischemic strokes, the hemorrhagic ones are listed under bleeding. We adjusted the table accordingly (341).

We studied your proposed paper. Unfortunately, this study refers to domestic primary retrievals of patients, focusing on primary care, which is not the topic of our paper.

  • Section 3.3. I would also be of interest to have the breakdown of injury type, such as head trauma. This would make it more relatable to other aeromedical services publications.

We added the breakdown of injury type to the table in the results, combined with age and gender data (341).

  • Other than mortality was any other follow-up conducted?
  • Cost section. There has been some aeromedical cost literature from Australia on mental health retrievals, however I agree the literature is lacking. This is a further strength of your paper.

Thank you very much. We highlighted this strength in an additional sentence (462).

  • Do you have severity codes? 1=high severity 2=moderate 3= low. Or does the triage categories cover clinical severity?

Our triage system STS covers severity codes from 1=high severity to 5=low severity (please see methods). We added some further information about the clinical presentation leading to the triage code for further clarity (175-178).

  • Would be interesting to have more information on diagnosis by severity, age, and gender, and mortality rates.

We absolutely agree with your comment. With the inclusion of a table to shorten the results section, we added the age, gender and mortality data by diagnosis (341). Since the severity of the diagnosis at arrival did not correlate with the initial illness or injury, we decided not to include this data.

Discussion

  • What does UNZ mean?

Universitäres Notfallzentrum (university emergency department). This has been changed to emergency department (of the Inselspital in Bern) (121,204, 420, 421).

  • I think much of this section requires more attention to the wider international literature. Specifically relating your findings to similar findings internationally.

We fully agree with your comment, yet it was not always possible to compare our results with existing literature. Nevertheless, we have tried to include existing literature wherever possible.

  • Statements like “people who travel to European and mostly neighbouring countries probably wish for medical treatment back in Switzerland due to the proximity. As the Swiss population is used to a very high quality of medical care, the fear of low quality and/or wrong treatment might be higher”, needs to be referenced, or acknowledged as a topic for future research. Did you collect any patient views on this?

  • Mortality section. You mention pre-existing comorbidities. Did you collect this information? Would be a very useful inclusion in the results.

We used the final emergency department (for outpatient or transferred treatment) or hospitalization diagnosis. Therefore, we could not analyse other comorbidities, as we did not have this additional data at our disposal.

  • Some literature from Australia.

We fully agree with your comment, yet it was not always possible to compare our results with existing literature. Not much international literature (including Australia) has been published regarding our topic. Where possible, we compared to existing studies.

Reviewer 2 Report

The authors investigate the topic of “Repatriations of ill and injured travelers and emigrants to Switzerland: a retrospective analysis at a tertiary emergency department from 2013-2018” They found that the number of repatriations has been continuously increasing in recent years and reached a mean value of more than one case per week at the Inselspital. It’s well written, but remains some concerns that are as follows:

  1. This study is a single center retrospective cross-sectional study describing 447 repatriations of ill and injured travelers transported back to Switzerland. The readers may be curious about what is the total number of repatriations of ill and injured travelers transported back to Switzerland? and also that how many percentage of 447 patients occupying of total repatriations patients during the study period? That means that authors must demonstrate certain evidence of 447 patients can be the representatives of the whole repatriation patients during study period.
  2. In result section, Costs, page 8, A multiple regression (table 5) including all variables,…. Being admitted as medical or surgical patient (GMR 0.79, 95% CI 0.68-0.93, p: 0.005). I just curious about the statistical analysis of that in table 4, the authors showed admission (injury and illness) was p=0.556, without statistical significance and it would be unreasonable why the parameter, admission, is picked up for multiple regression analysis and then generate a statistical significance (p=0.005). In addition, in table 5, the age, hospitalization, intensive care, surgery, respectively, all showed statistical significance, with geometric mean ratio > 1.0. These results mean that all the parameters would be positive correlated with the increased hospital costs. However, the geometer mean ratio of admission (injury vs. illness) is 0.79, < 1.0, that would be hard to understand and interpret.  

Author Response

Author's Reply to the Review Report (Reviewer 2)

Comments and Suggestions for Authors

The authors investigate the topic of “Repatriations of ill and injured travelers and emigrants to Switzerland: a retrospective analysis at a tertiary emergency department from 2013-2018” They found that the number of repatriations has been continuously increasing in recent years and reached a mean value of more than one case per week at the Inselspital. It’s well written, but remains some concerns that are as follows:

  1. This study is a single center retrospective cross-sectional study describing 447 repatriations of ill and injured travelers transported back to Switzerland. The readers may be curious about what is the total number of repatriations of ill and injured travelers transported back to Switzerland? and also that how many percentage of 447 patients occupying of total repatriations patients during the study period? That means that authors must demonstrate certain evidence of 447 patients can be the representatives of the whole repatriation patients during study period.

The share of repatriations to our clinic compared to all repatriations to Switzerland is about 6%. However, there are numerous hospitals in Switzerland that care for repatriated patients. We included absolute numbers from the REGA statistics regarding all repatriations to Switzerland in total (year 2018). Also, we have mentioned that the Inselspital in Bern is a major Swiss clinic that relevantly contributes to the medical care of repatriated patients. Nevertheless, we prefer not to report the absolute numbers, as this could weaken our study.

  1. In result section, Costs, page 8, A multiple regression (table 5) including all variables, Being admitted as medical or surgical patient (GMR 0.79, 95% CI 0.68-0.93, p: 0.005). I just curious about the statistical analysis of that in table 4, the authors showed admission (injury and illness) was p=0.556, without statistical significance and it would be unreasonable why the parameter, admission, is picked up for multiple regression analysis and then generate a statistical significance (p=0.005).

In addition, in table 5, the age, hospitalization, intensive care, surgery, respectively, all showed statistical significance, with geometric mean ratio > 1.0. These results mean that all the parameters would be positive correlated with the increased hospital costs. However, the geometer mean ratio of admission (injury vs. illness) is 0.79, < 1.0, that would be hard to understand and interpret. 

We were thinking that Surgery is a confounder regarding illness. If controlled for the confounder surgery in the multiple regression analysis, we can see that a patient, who has been admitted to the ED for an injury, generated significant less costs in comparison to patients, who were admitted due to illness. But we are absolutely open to exclude the type of admission from the multiple regression analysis, if recommended by you and especially, if this prevents confusion

Reviewer 3 Report

Thank you for the opportunity to review this article reporting on repatriations of ill and injured travellers and emigrants to Switzerland.

Overall, it is a very interesting study, well-written and well-designed. Yet, some parts need revision and modifications are suggested to improve the content. 

  • Introduction- Please add more details regard the common injuries/illness among travellers and emigrants (eg.,Siikamäki, H., Kivelä, P., Fotopoulos, M., Ollgren, J., & Kantele, A. (2015). Illness and injury of travellers abroad: Finnish nationwide data from 2010 to 2012, with incidences in various regions of the world. Eurosurveillance20(19), 21128.)
  • Introduction- I'm not sure that repatriations providers detail are necessary for this article
  • Introduction- please clarify: "... the number of repatriations to one major
    ...almost doubled in 12 years" - please add absoulte numbers 
  • Introduction- .."Furthermore, the data were compared
    to previous studies, particularly with respect to the characteristics of the patients"...Where this analysis appears? 
  • Figure 2- How did you calculate the total number? Is this sum of illness and injury?
  • Results- "About half of all patients were repatriated due to illness..and the other half suffered an injury.."Please move this sentence next to figure 2 
  • Results- "8 patients were initially treated in our ER intensive
    care room, as they were in a critical condition". Which condition? How you define critical condition?
  • Results- Why occupational groups is important for this study? Were they injured during their work? 
  • Results- Diagnostic group-, for example- What does it mean that patient admitted with myocardial infarction? Was the diagnosis made abroad or Inselspital? If the patient diagnosed abroad and received treatment for myocardial infarction, why he came to the hospital? please clarify and add an explanation regard the concept of repatriations 
  • Results- you need to shorten this section and present some of the result in a table 
  • DIscussion- In the current version the discussion section is a repetition of the results section, please revise and add details regarding previous studies/compare for other patients. For example, what is the mortality rate in Inselspital for "inhouse" patients? The authors should remember that this section is often considered the most important part of a research paper, because its formulate a deeper, more profound understanding of the research problem you are studying. 

Author Response

Author's Reply to the Review Report (Reviewer 3)

Comments and Suggestions for Authors

Thank you for the opportunity to review this article reporting on repatriations of ill and injured travellers and emigrants to Switzerland.

Overall, it is a very interesting study, well-written and well-designed. Yet, some parts need revision and modifications are suggested to improve the content.

  • Introduction- Please add more details regard the common injuries/illness among travellers and emigrants (eg.,Siikamäki, H., Kivelä, P., Fotopoulos, M., Ollgren, J., & Kantele, A. (2015). Illness and injury of travellers abroad: Finnish nationwide data from 2010 to 2012, with incidences in various regions of the world. Eurosurveillance, 20(19), 21128.)

This paper has been included (105), yet we wanted to keep our introduction at the existing length and decided not to give more detailed information about illnesses and injuries of travellers abroad.

  • Introduction- I'm not sure that repatriations providers detail are necessary for this article

This can be seen from different viewpoints (e.g. reviewer 1). We like to provide this basic amount of information for a better understanding of the patient population and our results.

  • Introduction- please clarify: "... the number of repatriations to one major

...almost doubled in 12 years" - please add absoulte numbers

We absolutely agree, the exact numbers have been added (108).

  • Introduction- .."Furthermore, the data were compared

to previous studies, particularly with respect to the characteristics of the patients"...Where this analysis appears?

We tried to compare our data to international publications, whenever possible. However, since we draw most of our comparisons to the 2000-2011 study from our hospital, we clarified this in the introduction (107-114).

  • Figure 2- How did you calculate the total number? Is this sum of illness and injury?

We used the total as well as the individual numbers for diseases and injuries.

  • Results- "About half of all patients were repatriated due to illness..and the other half suffered an injury.."Please move this sentence next to figure 2

This sentence has been moved (215-216).

  • Results- "8 patients were initially treated in our ER intensive

care room, as they were in a critical condition". Which condition? How you define critical condition?

By critical condition we meant that patients arrived with urgent triage category according to our triaging system or high staffing levels expected. This sentence has been extended for more clarity (224-225).

  • Results- Why occupational groups is important for this study? Were they injured during their work?

Since we did not have information on whether the patients were on vacation at the time of the accident/illness, or whether they were living abroad temporarily for work or as part of retired life, we added the information on occupation to allow a more comprehensive picture of the patients. Finally, we decided to remove this section completely after your recommendation in order to shorten the results section.

  • Results- Diagnostic group-, for example- What does it mean that patient admitted with myocardial infarction? Was the diagnosis made abroad or Inselspital? If the patient diagnosed abroad and received treatment for myocardial infarction, why he came to the hospital? please clarify and add an explanation regard the concept of repatriations.

We agree with your input of adding a concept of repatriations (we added this in our introduction 93-96)). Since most of the repatriated patients had been hospitalised abroad, they were mostly diagnosed abroad. More than 80% of all patients received further treatment in our clinic. For our study, we used the diagnoses from discharge reports for hospitalized patients or from emergency reports for outpatients and other transfers (178-180). Most patients were repatriated for further therapy, in some cases at the patients' request.

  • Results- you need to shorten this section and present some of the result in a table

All results have been revised, shortened and a table has been added (results).

  • Discussion- In the current version the discussion section is a repetition of the results section, please revise and add details regarding previous studies/compare for other patients. For example, what is the mortality rate in Inselspital for "inhouse" patients? The authors should remember that this section is often considered the most important part of a research paper, because its formulate a deeper, more profound understanding of the research problem you are studying.

We fully agree with your comment, yet it was not always possible to compare our results with existing literature. Nevertheless, we have tried to adapt the discussion section to your expectations.

Round 2

Reviewer 1 Report

Thank you for the opportunity to conduct a second review on this paper. All my comments have been addressed, and I have no further suggestions. 

The paper adds much to the aeromedical literature, and thankyou for your valuable work. 

I recommend publication. 

Reviewer 3 Report

I felt the authors addressed the above concerns very well and thoroughly in the revised paper